# ADABATCH: ADAPTIVE BATCH SIZES FOR TRAINING DEEP NEURAL NETWORKS

**Aditya Devarakonda**
Computer Science Division
University of California, Berkeley
`aditya@cs.berkeley.edu`

**Maxim Naumov & Michael Garland**
NVIDIA
Santa Clara, CA 95050, USA
`{mnaumov, mgarland}@nvidia.com`

## ABSTRACT

We introduce a new deep learning training approach that adaptively increases the batch size during the training process. Our method delivers the convergence rate of small, fixed batch sizes while achieving performance similar to large, fixed batch sizes. We train the VGG and ResNet networks on the CIFAR-100 and ImageNet datasets. Our results show that learning with adaptive batch sizes can improve performance by factors of up to 6.25 on 4 NVIDIA Tesla P100 GPUs while attaining similar accuracies to small batch sizes. Using our technique, we are able to train ImageNet with batch sizes up to $524,288$.

## 1 INTRODUCTION

Training deep neural networks with Stochastic Gradient Descent (SGD) typically uses a static batch size $r$, which is held constant throughout the training process. However, static batch sizes force the user to resolve an important conflict. On one hand, small batch sizes are desirable since they tend to produce convergence in fewer epochs (Das et al., 2016; Keskar et al., 2016). On the other hand, large batch sizes offer more data-parallelism which can improve computational efficiency and scalability (Goyal et al., 2017; You et al., 2017). Our approach to resolving this trade-off is to *adaptively increase* the batch size during training, beginning with an initial small batch size that increases between selected epochs (Devarakonda et al., 2017). For the experiments reported in this paper, we double the batch size at specific intervals and simultaneously adapt the learning rate $\alpha$ so that the ratio $\alpha/r$ remains constant. Our approach delivers the accuracy of training with small batch sizes, while improving performance during later epochs through the use of progressively larger batch sizes. Furthermore, the parallelism exposed by these large batches creates the opportunity for distributing work across many processors.

## 2 RELATING LEARNING RATE AND BATCH SIZE

Suppose that we have a weight matrix, $W_i$, at iteration $i$ during the training process and that $q$ iterations are required for one epoch of training (i.e., one pass over the data). After an epoch of training with a learning rate $\alpha$ and batch size $r$, the weight matrix update can be written as $W_{i+q} = W_i - \frac{\alpha}{r} \sum_{j=1}^{q} \Delta W_{i+j}$ with an update matrix $\Delta W_{i+j}$ computed at iteration $i + j$. Growing batches by a factor of $\beta > 1$ results in an effective batch size of $\beta r$ and epochs of $\tilde{q} = q/\beta$ iterations. This results in an update rule $W_{i+\tilde{q}} = W_i - \frac{\tilde{\alpha}}{\beta r} \sum_{j=1}^{\tilde{q}} \left( \sum_{k=1}^{\beta} \Delta W_{i'} \right)$, where index $i' = (j-1)\beta + k$. Notice that $W_{i+q}$ might be similar to $W_{i+\tilde{q}}$ only if we set the learning rate $\alpha = \tilde{\alpha}/\beta$ and assume that updates $\Delta W_i \approx \Delta W_{i'}$ are similar in both cases. This assumption was empirically shown to hold for fixed large batch size training with gradual learning rate warmup (Goyal et al., 2017) after the first few epochs of training. We can interpret $1/\beta$ as a learning rate decay. Thus, increasing the batch size can mimic learning rate decay, a relationship that Smith et al. (2017) have simultaneously emphasized. In our experiments, we increase batch sizes according to a fixed schedule.

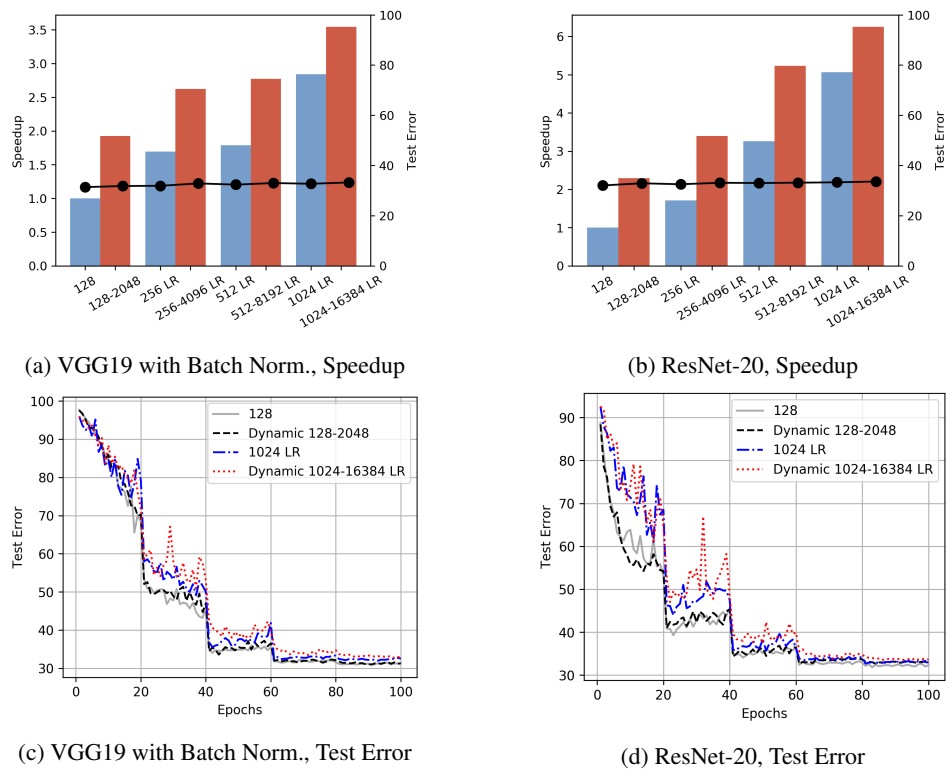

(a) VGG19 with Batch Norm., Speedup

(b) ResNet-20, Speedup

(c) VGG19 with Batch Norm., Test Error

(d) ResNet-20, Test Error

Figure 1: CIFAR-100 speedup (left vertical axis) and test errors (right vertical axis) for adaptive (in red) vs. fixed batch sizes (in blue), where "LR" uses learning rate warmup for the first 5 epochs.

## 3  EXPERIMENTAL RESULTS

We test our adaptive batch size technique using the VGG (Simonyan & Zisserman, 2014) and ResNet (He et al., 2016) deep learning networks on CIFAR-100 (Krizhevsky et al., 2009), and ImageNet (Deng et al., 2009) datasets. We benchmark our implementation[1] using PyTorch v0.1.12 running on 4 NVIDIA Tesla P100 GPUs interconnected via NVIDIA NVLink.

We begin our evaluation by training CIFAR-100 using VGG19 with Batch Normalization and ResNet-20. We use SGD with momentum of $0.9$ and weight decay of $5 \times 10^{-4}$. The baseline settings for both networks are fixed batch sizes of $128$, base learning rate of $0.1$, and learning rate decay by a factor of $0.25$ every $20$ epochs. The adaptive batch size experiments start with large initial batch sizes, perform gradual learning rate scaling over 5 epochs and double the batch every 20 epochs while decaying the learning rate by $0.5$. We perform 100 epochs of training for all settings. Figure 1 shows the speedups (left vertical axis) and test errors (right vertical axis) on (1a) VGG19 and (1b) ResNet-20. All speedups are normalized against the baseline fixed batch size of $128$. The additional "LR" labels on the horizontal axis indicate settings which require a gradual learning rate scaling in the first 5 epochs. Compared to the baseline fixed batch size setting, we see that adaptive 1024–16384 batch size attains average speedups (over 5 trials) of $3.54\times$ (VGG19) and $6.25\times$ (ResNet-20) with less than $2\%$ difference in test error. The test errors curves illustrate that adaptive batch sizes have similar behavior ($< 1\%$ difference) to their fixed batch size counterparts.

We also show the accuracy and convergence of AdaBatch on ImageNet training with the ResNet-50 network. Due to the large number of parameters, we are only able to fit a batch size of $512$ in multi-GPU memory. When training batch sizes $> 512$ we accumulate gradients. For example, when training with a batch size of $1024$ we perform two forward and backward passes with batch size $512$ and accumulate the gradients before updating the weights. For all experiments we train the ResNet-50 with a starting learning rate of $0.1$ and use learning rate warmup (Goyal et al., 2017). We

---

[1] https://github.com/NVlabs/AdaBatch

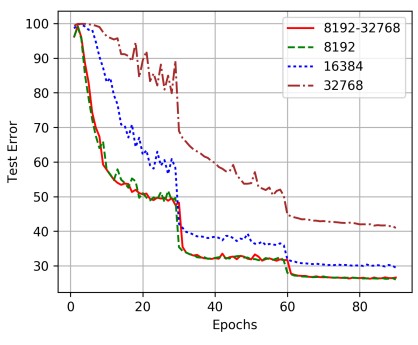

(a) ResNet-50, starting batch size 8192.

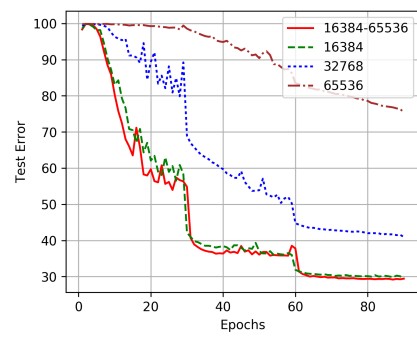

(b) ResNet-50, starting batch size 16384.

Figure 2: ImageNet test errors curves for adaptive versus fixed batch sizes with LR warmup.

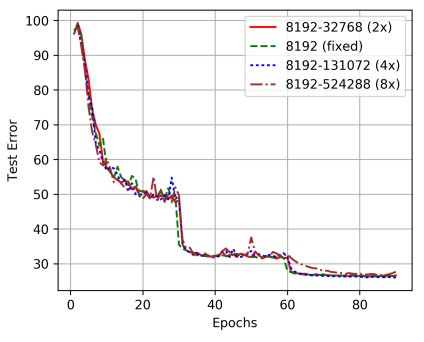

(a) ResNet-50, starting batch size 8192.

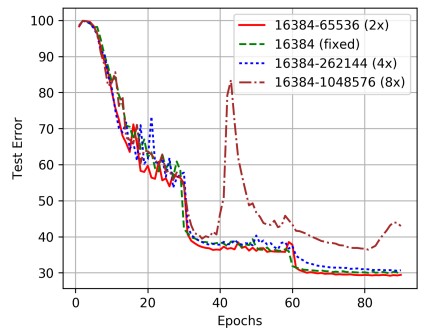

(b) ResNet-50, starting batch size 16384.

Figure 3: Comparison of ImageNet test errors curves for adaptive batch sizes with LR warmup and batch size increases of 2x, 4x, and 8x.

then use a learning rate decay of $0.1$ every 30 epochs for fixed batch size experiments. For adaptive batch sizes, we simultaneously double the batch size and decay the learning rate by $0.2$ every 30 epochs. The network is trained using SGD with momentum of $0.9$ and weight decay of $10^{-4}$.

Figure 2 illustrates the test errors of large batch size training on ImageNet. We use a baseline batch size of 256 for learning rate warmup. Our results indicate that adaptive batch size convergence is similar to small, fixed batch size convergence and superior to large, fixed batch size. In Figure 3, we explore the convergence behavior of AdaBatch when batch size is increased by factors of $2\times$, $4\times$ and $8\times$ and learning rates decayed by $0.2$, $0.4$ and $0.8$, respectively, every 30 epochs. All other training parameters are fixed. Our results indicate that the test error curves are comparable for a wide range of increase factors. Notably, AdaBatch enables ImageNet training with batch sizes of up to $524,288$ without significantly altering test error. In Figure 3b, increasing the batch size by $8\times$ results in poor convergence. This is a result of increasing the batch size too much and too early during training. It is important to tune the increase factor proportional to the starting batch size.

## 4 CONCLUSION

In this paper we have developed an adaptive scheme that dynamically increases the batch size during training. We have shown that by using our scheme to train CIFAR-100 and ImageNet on the VGG and ResNet networks, we can maintain the better test accuracy of small batches, while obtaining higher performance often associated with large batches. Our results demonstrate that AdaBatch can attain speedups of up to $6.25\times$ on 4 NVIDIA P100 GPUs and that ImageNet can be trained with batch sizes of up to $524,288$ with less than a 1% change in accuracy.

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
