# OpenReview forum: "AdaBatch: Adaptive Batch Sizes for Training Deep Neural Networks"
_ICLR.cc/2018/Workshop — Reject_

### Official Review · AnonReviewer1 · 2018-03-09
**Nice idea, but it's been done before**

**Rating:** 4
**Confidence:** 4

**Review:**

This paper proposes dynamically increasing the minibatch size on a fixed schedule, since at early iterations noisy gradient estimates are fine, since the noise is smaller relative to the potential improvement, whereas in later iterations noise has a bigger impact. This makes the earlier iterations faster, potentially speeding up convergence, which they demonstrate experimentally. This is a nice, intuitive idea, and I like the simplicity of performing the increase on a fixed schedule.

However, the KDD 2017 paper "Small Batch or Large Batch? Gaussian Walk with Rebound can Teach", by Yin, Luo and Nakamura, proposes the same idea, except that they dynamically increase the batch size based on a heuristic (for which they give some theoretical justification). It may be that a fixed schedule is better--it's certainly simpler--but I think that a comparison must be made to this previous work.

---

### Official Review · AnonReviewer2 · 2018-03-09
**Good paper with presentational issues**

**Rating:** 7
**Confidence:** 4

**Review:**

The authors present a clean method (double step size and batch size when otherwise decreasing step size) that works well.

There are some presentational / writing issues.

In figure 1, I'd like to see the test error values in a table, not as giant black circles. I can't figure out the test error of the baseline without holding a ruler to my screen. There's enough whitespace around those charts to fit in a table without increasing the paper length.

I find your formulas in section 2 confusing / nonsensical.
- What is the update matrix \Delta W? Either define this matrix or write the update equation in terms of gradients directly.
- Why does a single large batch iteration compute a sum of \beta update matrices? The large-batch method doesn't know its batch size was recently increased by a factor \beta, it only knows its current batch size. Hence, the update scheme for large-batch should not depend on \beta.
- I think the i' subscripts make no sense. In the original definition of W_{i+q}, the subscript of the update matrices i+j ranges from i to i+q. But in the definition of W_{i+\tilde{q}}, the indices of the update matrices are independent of i and range from 0 to q? Something seems to be going wrong here.
- The statement \Delta W_i \approx \Delta W_{i'} also makes no sense to me. i is a fixed subscript indicating the epoch. i' is a running subscript indicating individual iterations. Why are you comparing the two?

Your method can be summed up simply as follows:

"Whenever we would usually decay the learning rate by factor k, instead we decay it by k/c and increase the batch size by c."

I think a sentence like this that lets the reader know what you are doing should appear in section 2. I had to re-read section 3 multiple times to realize that this is what you are doing. Your method is essentially only specified implicitly via your desciprtion of the experimental protocol in section 3.

---

### Official Review · AnonReviewer3 · 2018-03-10
**Not any new insights**

**Rating:** 4
**Confidence:** 4

**Review:**

The relationship between changing batch sizes and learning rate decay is well understood as pointed out by authors themselves. And the results shown in the plots show typical patterns. Although a valid study and correct idea, the paper does not offer new insights into the working of sgd or otherwise.

---

### Decision · Program_Chairs · 2018-03-20
**ICLR 2018 Workshop Acceptance Decision**

**Decision:**

Reject

**Comment:**

Based on the reviews, this paper has not been accepted for presentation at the ICLR workshop. However, the conversation and updates can continue to appear here on OpenReview.